# Efficacy of Quetiapine Monotherapy and Combination Therapy for Patients with Bipolar Depression with Mixed Features: A Randomized Controlled Pilot Study

**DOI:** 10.3390/ph16020287

**Published:** 2023-02-14

**Authors:** Zheng Wang, Danhua Zhang, Yanli Du, Yin Wang, Tingting Huang, Chee H. Ng, Huimin Huang, Yanmeng Pan, Jianbo Lai, Shaohua Hu

**Affiliations:** 1Department of Psychiatry, The First Affiliated Hospital, Zhejiang University School of Medicine, Hangzhou 310003, China; 2Department of Psychiatry, Dongyang People’s Hospital, Jinhua 322103, China; 3Clinical Trial Unit, Children’s Hospital of Fudan University, National Children’s Medical Center, Shanghai 310005, China; 4Affiliated Hangzhou First People’s Hospital, Zhejiang University School of Medicine, Hangzhou 310003, China; 5Department of Psychiatry, The Melbourne Clinic and St Vincent’s Hospital, University of Melbourne, Richmond, VIC 3121, Australia; 6Brain Research Institute, Zhejiang University, Hangzhou 310003, China; 7Zhejiang Engineering Center for Mathematical Mental Health, Hangzhou 310003, China; 8Department of Neurobiology, NHC and CAMS Key Laboratory of Medical Neurobiology, School of Brain Science and Brian Medicine, and MOE Frontier Science Center for Brain Science and Brain-Machine Integration, Zhejiang University School of Medicine, Hangzhou 310003, China

**Keywords:** bipolar depression with mixed features, quetiapine, valproate, lithium, monotherapy, combination therapy

## Abstract

Effective pharmacotherapy of bipolar depression with mixed features defined by DSM-5 remains unclear in clinical treatment guidelines. Quetiapine (QTP) and valproate have potential treatment utility but are often inadequate as monotherapy. Meanwhile, the efficacy of combination therapies of QTP plus valproate or lithium have yet to be verified. Hence, we conducted a randomized controlled pilot study to evaluate the efficacy of QTP monotherapy in patients with bipolar depression with mixed features defined by DSM-5 and compared the combination therapy of QTP plus valproate (QTP + V) versus QTP plus lithium (QTP + L) for those patients who responded insufficiently to QTP monotherapy. Data was analyzed according to the intent-to-treat population. Generalized linear mixed model was performed by using “nlme” package in R software. A total 56 patients were enrolled, among which, 35 patients responded to QTP alone, and 11 and 10 patients were randomly assigned to QTP + V and QTP + L group, respectively. Nearly 60% enrolled patients responded to QTP monotherapy at the first two weeks treatment. No statistically significant difference in efficacy between QTP + V and QTP + L was observed. In conclusion, QTP monotherapy appeared to be efficacious in patients with bipolar depression with mixed features, and for those who responded insufficiently to QTP, combining with either valproate or lithium appeared to have positive effects.

## 1. Introduction

Bipolar disorder is a severe chronic psychiatric disorder which presents with recurrent manic or hypomanic and depression episodes resulting in prominent impaired social functioning. The etiology of bipolar disorder has not yet been elucidated, however, several risk factors including genetic [1], oxidative stress [2], neuroimmune [3], gut microbiome [4] have been found to play critical roles. There has been growing interest in the phenomenology and clinical implications of ‘mixed states’ in bipolar disorder—that is, the simultaneous occurrence of both manic/hypomanic and depressive symptoms in an individual [5]. Diagnostic and Statistical Manual of Mental Disorders 5th edition (DSM-5) replaced the DSM-IV mixed episode diagnosis with the “mixed features”, defined by the presence of at least three non-overlapping opposite-pole symptoms in the context of a syndromal depressive, hypomanic, or manic episode. The rate of diagnosis of bipolar disorder with mixed features, the methods used to treat this subtype, and clinical studies on mood disorders will probably be impacted by these changes, which reflect a more lenient application of the “with mixed features” specifier to both polarities of bipolar disorder [6]. Bipolar depression with mixed features has been identified in 21–76% of depressed patients [7,8,9,10], and has been associated with adverse clinical outcomes such as younger onset age [8,9], suicidal behavior [11,12], unstable or severe course of illness [6,8], and greater functional disability [13]. Thus, there is an urgent need to identify suitable treatment for patients with bipolar depression with mixed features.

Clinical reports indicate that conventional antidepressants may be ineffective for patients with bipolar depression with mixed features and may be related to potential treatment-related complications like suicidal ideation and behavior, agitation and manic switch [14]. Atypical antipsychotics and mood stabilizers have been recommended as therapy alternatives [15,16], but there is scant experimental evidence to support these treatment options. The available research evidence for pharmacological treatment of bipolar depression with mixed features is limited to a few studies using DSM-5 criteria. Recently, the Canadian Network for Mood and Anxiety Treatments (CANMAT) and International Society of Bipolar Disorders (ISBD) 2021 guidelines stated that no agents have sufficient evidence for first line treatment of DSM-5 bipolar depression with mixed features [17]. Given the limitations of current pharmacological treatment, there is a pressing need for new treatment options for this condition.

Evidence shows that second generation antipsychotics appear to offer favorable improvements in Montgomery Montgomery-Åsberg Depression Rating Scale (MADRS) and Young Mania Rating scale (YMRS) scores compared to placebo in the treatment of patients with acute bipolar depression with mixed features [18]. Cariprazine and lurasidone are recommended by CANMAT and ISBD 2021 as second line options [17]. Based on the efficacy in treating acute bipolar depression [19], mania [20] and in preventing mood episodes in those with index mixed episode [21], quetiapine (QTP) is considered to have utility for both poles in those with depression plus mixed features, both as monotherapy and in combination with other mood stabilizers [17]. Quetiapine is a partial agonist of 5-HT1A receptor, and it also has a certain antagonistic effect on serotonin 5-HT2A and dopamine D2 [22]. The affinity for different 5-HT receptors appears to play an important role in the characteristic of mood stabilizing [23]. In addition, QTP is particularly effective on insomnia, psychomotor agitation, elevated mood, racing thoughts, irritability, impulsivity, pressured speech, hypersexuality and psychosis, which may contribute to reduce the clinical risks during the episode of bipolar depression with mixed features [24]. RCTs on second generation antipsychotics for bipolar depressive episodes with mixed features in recent years have been focused on lurasidone [25,26]. There is still a lack of controlled studies on the efficacy of QTP in the treatment of patients with bipolar depression with mixed features defined by DSM-5.

Lithium and valproate as mood stabilizers are standard treatment choices for the management of bipolar disorder. As a mood stabilizer, studies on valproate have found efficacy in treating depression [27], manic symptoms in mania and mania with mixed features [28]. Valproate is recommended by CANMAT and ISBD 2021 guideline as third line option for bipolar depressive episode with mixed features. On the other hand, lithium is the most effective drug for the treatment of classical recurrent depressive and bipolar disorders and is the only drug that has shown a clear effect in preventing suicide independently from diagnosis [29]. For bipolar depressive episode, lithium was recommended as first line option in CANMAT and ISBD 2018 guideline [27], the third line option in World Federation of Societies of Biological Psychiatry (WFSBP) 2010 guideline [30], and the fourth line option in International College of Neuro-Psychopharmacology (CINP) 2017 guideline [31]. However, no recommendations were given for lithium to treat the acute mixed state (defined by DSM-IV) in CANMAT and ISBD 2021 [17], WFSBP 2018 [32], CINP 2017 [31], British Association for Psychopharmacology (BAP) 2016 guideline [33], National Institute for Health and Care Excellence (NICE) 2014 [34]. One critical point in the CANMAT and ISBD 2021 guidelines is that no recommendation could be made for the clinical use of lithium in the management of depression with mixed features [17]. Although it is not surprising, based on the known lack of studies on the role of lithium in treating patients with DSM-5-defined bipolar depression with mixed features. High-quality trials are needed to support the evidence-based treatment with lithium in this episode [35,36].

The primary goal of treatment for a patient with bipolar depression with mixed features is to achieve symptomatic recovery with stable mood as quickly as possible. In clinical practice, many patients with bipolar disorder need combination treatments with two or possibly more agents to achieve response. Studies have found that combined treatments with an atypical antipsychotic plus a mood stabilizer are more effective with respect to monotherapies in achieving clinical stabilization of bipolar patients [37]. Similarly, treatment guidelines have recommended initiating treatment that is based on the best evidence for efficacy and tolerability, while second generation antipsychotics in combination with mood stabilizer should be reserved for more severe presentations as first-line choice or as a subsequent step when another first-line medication failed. However, the efficacy of combination therapies of QTP plus valproate or lithium have yet to be verified and the choice between these two therapeutic options is usually not immediate in clinical practice. Thus, we conducted a randomized pilot trail to investigate the effectiveness of monotherapy of QTP for patients with bipolar depression with mixed features, and compare combination treatment of QTP plus valproate versus QTP plus lithium for those who have insufficient response to QTP.

## 2. Results

### 2.1. Participants

A total of 65 patients were screened for eligibility and 56 patients who met inclusion and exclusion criteria were recruited (N = 5 did not meet entry criteria and N = 4 withdrew consent). Figure 1 details the patient disposition during 2 treatment phases. Baseline characteristics of all patients were summarized in Table 1. Of 56 enrolled patients, 35 (62.5%, 35/56) patients responded to QTP; 21 (37.5%, 21/56) patients with MARDS score reduction of less than 20% were randomly assigned to either valproate group (11 patients, treatment: QTP + V) or lithium group (10 patients, treatment: QTP + L). Of the total number who were randomized, 11 patients (52.4%, 11/21) completed the 6 weeks treatment, while 12 patients (34.3%, 12/35) completed the 6 weeks treatment in the QTP group (Figure 1). The mean (±SD) age, education and illness duration for the participants were 21.80 ± 7.05 years, 12.33 ± 2.34 years and 63.41 ± 55.41 months, respectively. The characteristics of the participants at baseline were similar in the QTP, QTP + V and QTP + L groups (Table 1). The total score of MADRS, HAMA, YMRS and CUDOS for the patients at the enrolment were 29.75 ± 8.07, 26.95 ± 6.43, 13.54 ± 5.86 and 18.49 ± 7.76, respectively. After 2 weeks treatment, the scores of MARDS, HAMA, YMRS and CUDOS were 26.70 ± 9.81, 28.89 ± 8.27, 7.20 ± 5.98 and 10.50 ± 7.88, respectively in QTP + V group. For QTP + L group, the MARDS score was 26.78 ± 7.46, HAMA score was 28.44 ± 9.13, YMRS score was 11.33 ± 7.63 and CUDOS score was 14.11 ± 9.98. And in QTP group, the MARDS score was 13.15 ± 6.68, HAMA score was 19.88 ± 11.43, YMRS score was 6.06 ± 4.85 and CUDOS score was 9.97 ± 6.67 at week 2. At end of week 8, the scores of MARDS, HAMA, YMRS and CUDOS were reduced to 17.00 ± 7.07, 13.50 ± 9.54, 5.00 ± 4.24, 6.00 ± 7.35 in QTP + V group and 19.86 ± 9.84, 19.57 ± 9.48, 7.43 ± 7.18, 10.29 ± 4.89 in QTP + V group respectively. And in QTP group, the MARDS score was 6.92 ± 7.05, HAMA score was 10.08 ± 9.47, YMRS score was 3.17 ± 2.52 and CUDOS score was 5.75 ± 5.07 at week 8 (Table 2).

### 2.2. Change of the Scores

No significant differences in changes of MADRS, HAMA, CUDOS and YMRS scores were observed between QTP + V and QTP + L group over the 6 weeks with nonsignificant group by time interaction effect (MADRS, *p* = 0.545; HAMA, *p* = 0.320; CUDOS, *p* = 0.640; YMRS, *p* = 0.300) (Table 3). Within group analysis showed that significant reduction of MADRS (mean difference −7.18 (−13.04 to −1.32), *p* = 0.025), HAMA (mean difference −9.25 (−16.29 to −2.21), *p* = 0.017) and YMRS (mean difference −4.82 (−9.33 to −0.31), *p* = 0.047) scores after 6 weeks treatment in QTP + L group. For QTP + V group, significant reduction was only observed in MADRS (mean difference −8.6 (−15.75 to −1.45), *p* = 0.027) and HAMA (mean difference −16.25 (−24.93 to −7.57), *p* < 0.001) scores (Table 3). All measures in QTP + L group showed a downward trend, however, in QTP + V group, the downward trend of the change in MADRS scores went into reverse at the second visit and started to creep up. For YMRS scores, the trend was steady until the week 4 and raised at week 8. For HAMA scores, the scores in the QTP + V group went down sharply after randomization and point estimation became the lowest at the last visit (QTP + V:13.50 vs. QTP + L:19.57) (Figure 2). Similar results were found when the QTP group was included into comparison analysis (Figure 3). For the YMRS and CUDOS scores, QTP group showed lower point estimation when compared with other two groups. For MADRS scores, except for the QTP + V group which showed a reversed trend, the QTP and QTP + L groups showed a downwards trend as the trial progressed.

## 3. Discussion

To our knowledge, this pilot study is the first study to examine the monotherapy effect of QTP in patients with bipolar depression with mixed features defined by DSM-5, and compare the effect of combination therapy of QTP plus valproate versus QTP plus lithium in patients who responded insufficiently to QTP after the first two weeks treatment. Nearly 60% eligible patients responded to monotherapy QTP with the first treatment. No significant differences were subsequently observed between QTP + V and QTP+ L groups in those who were insufficient responders. However, within group comparison showed that the total scores of MADRS, HAMA and YMRS were significant lower over the 6 weeks treatment in QTP + L group whereas only the MADRS and HAMA scores were significant reduced in QTP + V group. The total score of MADRS, HAMA, CUDOS and YMRS showed different trends among the groups.

Several studies showed that QTP is effective in treating bipolar disorder patients in depressive episode [19]. As far as we know, no study has evaluated the monotherapy efficacy of QTP in treating of bipolar depression with mixed features defined by DSM-5. Our study demonstrated that, QTP monotherapy appeared to have an efficacious effect (62.5%) in reducing total MARDS scores in patients with bipolar depression with mixed features within two weeks. There is little data for the efficacy of lithium in treating patients with bipolar depression with mixed features so far. A review article suggested that QTP monotherapy was found to be more efficacious than lithium in treating acute bipolar depression [38]. However, research on treatment for bipolar depression with mixed features patients with insufficient response to monotherapy is lacking. As DSM-5 suggested, the new classification of bipolar depression with mixed features may highlight a new consideration for treatment of this clinical subgroup. Our study demonstrated that the point estimations of four outcome measures in the QTP group were all lower than QTP + L group, which may suggest that the initial response to QTP may have a substantial contribution for the further therapy.

Due to the proven efficacy in treating depression [27] and mania with mixed features [28], divalproex was expected to be an effective adjunctive therapy for those patients who had poor response to QTP. However, significant reduction of scores were observed in four measures only in QTP + L group. Meanwhile, point estimations of the efficacy in QTP + L group in four outcome measures were all lower than that in QTP + V group at the week 8. This may suggest that compared with QTP plus valproate, QTP plus lithium could be a suitable treatment option for bipolar depression with mixed features patients when QTP shows inadequate response. However, we should interpret this result carefully due to the single arm design and small sample size.

Emerging evidence suggests that patients with bipolar depression with mixed features are a special clinical population, characterize not only by depressive symptoms, but also by irritability, rapid cycling, suicidality and poor treatment response [39,40]. Early clinical stabilization remains a clinical priority. Clinical trial results imply that clinicians can be confident in predicting when a therapy is not effective during short-term treatment since the absence of early improvement seems to be a highly reliable predictor of subsequent non-response. Hence, the measure of at least 20% improvement within the first two weeks was selected because it has been noted to be a clinically meaningful shift that is simple for clinicians to recognize. A change in therapy may be beneficial for patients who do not show signs of improvement within the first two weeks of treatment [41]. Combined treatments should be reserved for patients who do not achieve clinical stabilization with monotherapy [37]. In line with these considerations, the current study suggested that QTP monotherapy was efficacious in patients with bipolar depression with mixed features, and for those who failed to reach 20% improvement in depressive symptoms within first weeks adding lithium or valproate to ongoing QTP could be a useful next step.

Several limitations of the present study need to be considered when interpreting the findings. Firstly, the sample size is small, which may limit the power to find the difference between two groups, even though down trends were showed in QTP +L group when compared with QTP +V group. The result was limited in its generalizability due to the small sample. Replication with a larger sample from more centers is needed. Secondly, the side effects were not recorded. The treatment was generally well tolerated in all patients through the trial. Thirdly, the plasma levels of valproate/lithium were not measured. In addition, the higher dropout rates in QTP + V group may have violated the missing at random assumption of GLMM and increased Type II error in a potentially significant comparison.

## 4. Conclusions

In conclusion, there is an urgent need for evidence-based therapies for patients with bipolar depression with mixed features. Quetiapine monotherapy appeared to show efficacy in treating patients with bipolar depression with mixed features. For those who responded insufficiently to QTP, combining with either valproate or lithium appeared to have beneficial effects. A larger sample size clinical trial is warranted in future studies.

## 5. Materials and Methods

### 5.1. Patients

Study enrolled outpatients aged 18 to 65 years who met DSM-5 diagnostic criteria for bipolar disorder type I or II and were required to have a current major depressive episode with mixed features, with a score ≥20 on the MADRS and a YMRS score of ≤19 at both screening and baseline. Patients were required to have two or three of the following manic symptoms, on most days of the current episode of depression: elevated or expansive mood, inflated self-esteem or grandiosity, more talkative than usual or pressure to keep talking, flight of ideas or racing thoughts, increase in energy or goal-directed activity, increased or excessive involvement in activities with a high potential for negative consequences and decreased need for sleep. Eligible patients were not taking any psychotropics for 4 consecutive weeks prior to the enrollment. Patients were excluded if they met DSM-5 criteria for a current or lifetime diagnosis of schizophrenia and related psychotic disorder, borderline personality disorder, substance use disorder within the past 6 months, intellectual disability and autism spectrum disorder. Patients who had psychotic features were excluded. Additional exclusion criteria included a current serious suicidal or homicidal risk, or a suicide attempt within the past 6 months; a clinically significant medical illness; or any clinically significant findings on laboratory tests or electrocardiogram (ECG); undergoing electroconvulsive therapy (ECT) within the past 3 months; contraindication to any study drugs; participation in an investigational drug trial within 30 days before the start of the trial; pregnant and lactating women; or women of childbearing potential who were without adequate contraception.

### 5.2. Study Design

This was a randomized, eight-week pilot trial of monotherapy of QTP in patients with bipolar depression with mixed features, comparing the combination therapy of QTP plus valproate (QTP + V) versus QTP plus lithium (QTP + L) for those with MADRS total score reduction less than 20% after first two weeks treatment of QTP. Eligible patients initially received QTP monotherapy for 2 weeks. At the end of week 2, adequate responders (defined as achieving a ≥20% improvement compared with baseline in MARDS total score) were required to continue on QTP monotherapy, while the inadequate responders were randomly assigned in a 1:1 ratio to the groups of QTP +V and QTP +L for 6 weeks treatment. The randomization list was computer generated and none of the investigators or patients had access to the list.

The study employed a flexible dosing schedule for QTP, valproate and lithium. Guidance for the dosage of QTP involved increasing the first day’s dose (100 mg/day) by 100 mg on every other day until day 3 (300mg/day), and then adjusting the dose ranged 300 mg/day to 600 mg/day according to clinical symptoms and tolerability. Lithium was started from 300 mg/day and increased to 600–900 mg/day within 3 days, and valproate was started from 500 mg/day and adjusted to 1000 mg/day within 3–5 days according to clinical symptoms and tolerability. The following were not permitted during the trial: antipsychotics other than QTP; mood stabilizers other than lithium or valproate; antidepressants; and noninvasive brain stimulation techniques such as repetitive transcranial magnetic stimulation (rTMS). Treatment with anticholinergic agents or propranolol was permitted as needed.

Clinical measures included: MARDS as the primary outcome and Hamilton Rating Anxiety Scale (HAMA), YMRS and Clinically Useful Depression Outcome Scale supplemented with questions for the DSM-5 mixed features specifier (CUDOS-M) as secondary outcomes. All the scales were evaluated by senior psychiatrists on baseline, week 2, week 6 and week 8, respectively. Throughout the trial, the same psychiatrist conducted all ratings for an individual patient.

### 5.3. Statistical Analyses

A prior sample size estimation indicated that 37 participants per arm were required to obtain 80% power for detecting mean difference of 4 points (SD = 6) of MADRS scores after 6 weeks intervention, using two-tailed test with an alpha level of 0.05.

The outcomes (primary: change from randomization to week 6 in total MADRS score; secondary: change from randomization to week 6 in total scores, including HAMA, YMRS and CUDOS were repeated measured continuous variable and were analyzed according to the intent-to-treat (ITT) population by using Generalized linear mixed model. Generalized linear mixed mode was performed using maximum likelihood (ML) method with group and week and interaction of group with week as fixed effect, patient ID as random effect, age and sex as explanatory variables. Gaussian distribution and identity link function was used in the Generalized linear mixed mode model.

Due to the small sample size, missing values were not processed with multi-imputation. Generalized linear mixed models use listwise deletion of individual observations rather than entire persons and maximum likelihood estimation to robustly handle data that are missing at random or completely at random [42].

Continuous variables were presented as the mean ± standard deviation (SD) or interquartile range (IQR) as appropriate, and categorical variables were expressed as absolute numbers and percentages. According to the data distribution, t-test or Wilcoxon test would be used as appropriate. A two-side *p* value of less than 0.05 was considered to indicate statistical significance for the outcomes. All the analysis was performed in R software (version 4.1).

## Figures and Tables

**Figure 1 pharmaceuticals-16-00287-f001:**
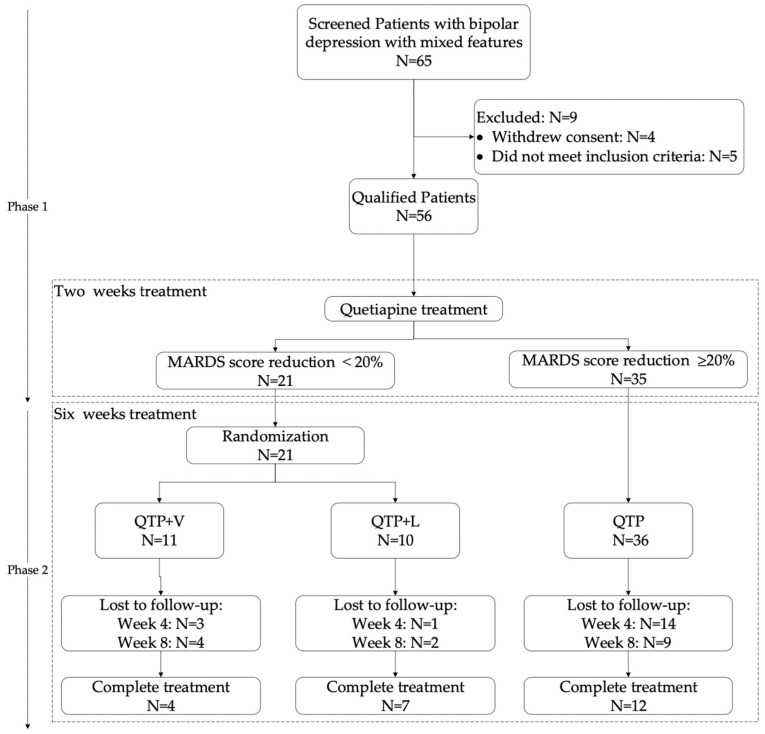
Enrolment and randomization.

**Figure 2 pharmaceuticals-16-00287-f002:**
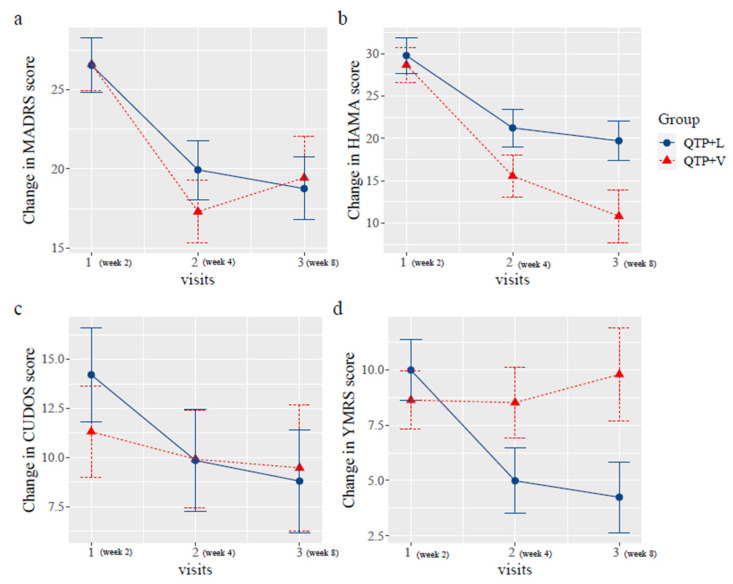
Change of MADRS, HAMA, CUDOS-M and YMRS scores of QTP + L and QTP + V group during six weeks treatment. QTP + V: quetiapine plus valproate group; QTP + L: quetiapine plus lithium group; analyses were conducted with the use of a generalized linear mixed model, treating group as fixed effect, patient ID as random effect, time * group as interaction and age, sex as explanatory variables.

**Figure 3 pharmaceuticals-16-00287-f003:**
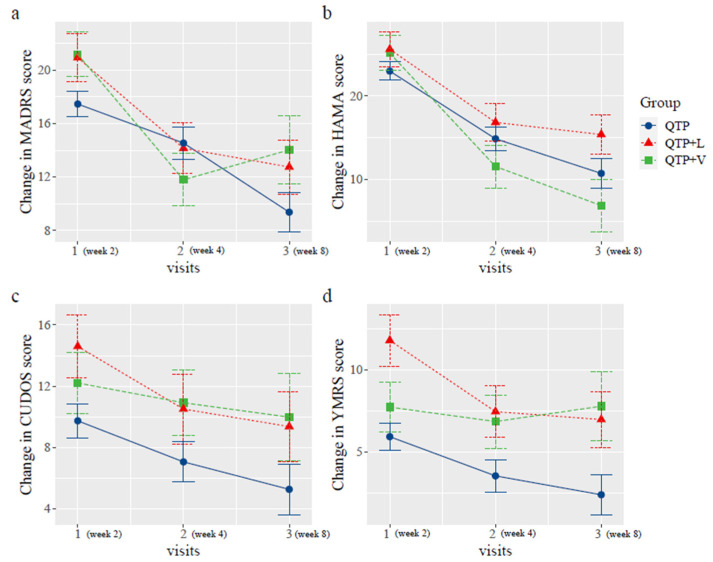
Change of MADRS, HAMA, CUDO-MS and YMRS scores of QTP, QTP + L and QTP+S during six weeks treatment. QTP: quetiapine group; QTP + V: quetiapine plus valproate group; QTP + L: quetiapine plus lithium group; analyses were conducted with the use of a generalized linear mixed model, treating group as fixed effect, patient ID as random effect, time * group as interaction and age, sex as explanatory variables.

**Table 1 pharmaceuticals-16-00287-t001:** Characteristics of the enrolled patients *.

	QTP + V (N = 11)	QTP + L (N = 10)	QTP (N = 35)	Total (N = 56)
**Age**	20.18 ± 4.42	22.40 ± 8.30	22.14 ± 7.44	24.80 ± 7.05
**Sex (N/%)**				
**Male**	3 (27.27)	4 (40.00)	18 (51.43)	25 (44.64)
**Female**	8 (72.73)	6 (60.00)	17 (48.57)	31 (55.35)
**Education (year)**	11.50 ± 2.37	11.80 ± 2.30	12.71 ± 2.32	12.33 ± 2.34
**Illness Duration (month)**	62.40 ± 55.37	55.50 ± 36.60	66.03 ± 60.85	63.41 ± 55.4

* QTP + V: quetiapine plus valproate group; QTP + L: quetiapine plus lithium group; QTP: quetiapine group; Plus-minus values are means ± SD.

**Table 2 pharmaceuticals-16-00287-t002:** Total scores of HAMA, YMRS, MARDS, CUDOS of three groups at enrolment, two weeks and eight weeks (Mean ± SD).

		QTP + V (N = 11)	QTP + L (N = 10)	QTP (N = 35)	Total (N = 56)
**Baseline**	**MARDS**	N = 11	N = 10	N = 35	N = 56
		30.09 ± 8.11	33.20 ± 7.77	28.66 ± 8.08	29.75 ± 8.07
	**HAMA**	N = 11	N = 10	N = 35	N = 56
		25.27 ± 10.55	30.50 ± 9.58	26.46 ± 9.05	26.95 ± 6.43
	**YMRS**	N = 11	N = 10	N = 35	N = 56
		14.55 ± 6.83	14.20 ± 6.00	13.03 ± 5.62	13.54 ± 5.86
	**CUDOS**	N = 11	N = 8	N = 30	N = 49
		16.91 ± 6.47	17.88 ± 5.33	19.23 ± 8.76	18.49 ± 7.76
**Week 2**	**MARDS**	N = 10	N = 9	N = 34	N = 53
		26.70 ± 9.81	26.78 ± 7.46	13.15 ± 6.68	18.02 ± 9.84
	**HAMA**	N = 9	N = 9	N = 33	N = 51
		28.89 ± 8.27	28.44 ± 9.13	19.88 ± 11.43	22.98 ± 11.22
	**YMRS**	N = 10	N = 9	N = 34	N = 53
		7.20 ± 5.98	11.33 ± 7.63	6.06 ± 4.85	7.17 ± 5.82
	**CUDOS**	N = 10	N = 9	N = 32	N = 51
		10.50 ± 7.88	14.11 ± 9.98	9.97 ± 6.67	10.08 ± 7.56
**Week 4**	**MARDS**	N = 8	N = 9	N = 21	N = 38
		16.25 ± 6.73	22.00 ± 10.71	10.38 ± 6.86	14.37 ± 9.10
	**HAMA**	N = 8	N = 9	N = 21	N = 38
		15.88 ± 8.32	21.3 ± 9.70	11.81 ± 6.87	14.92 ± 8.64
	**YMRS**	N = 8	N = 9	N = 21	N = 38
		5.75 ± 5.97	8.00 ± 5.74	4.10 ± 2.84	5.37 ± 4.57
	**CUDOS**	N = 8	N = 7	N = 21	N = 36
		10.63 ± 8.75	12.00 ± 8.14	7.38 ± 4.40	9.00 ± 6.46
**Week 8**	**MARDS**	N = 4	N = 7	N = 12	N = 23
		17.00 ± 7.07	19.86 ± 9.84	6.92 ± 7.05	12.61 ± 9.80
	**HAMA**	N = 4	N = 7	N = 12	N = 23
		13.50 ± 9.54	19.57 ± 9.48	10.08 ± 9.47	13.57 ± 9.99
	**YMRS**	N = 4	N = 7	N = 12	N = 23
		5.00 ± 4.24	7.43 ± 7.18	3.17 ± 2.52	4.78 ± 4.83
	**CUDOS**	N = 4	N = 7	N = 12	N = 23
		6.00 ± 7.35	10.29 ± 4.89	5.75 ± 5.07	7.17 ± 5.58

**Table 3 pharmaceuticals-16-00287-t003:** Changes of MADRS, HAMA, CUDOS-M and YMRS scores in QTP + L and QTP + V group between week 2 and week 8 and interaction part of the model estimated by generalized mixed model ^#^.

		Week 2	Week 8	Difference (95%CI)	*p* Value
**MADRS**	time * group				0.545
	QTP + L	26.78	19.86	−7.18 (−13.04, −1.32)	0.025
	QTP + V	26.70	17.00	−8.6 (−15.75, −1.45)	0.027
**HAMA**	time * group				0.320
	QTP + L	28.44	19.57	−9.25 (−16.29, −2.21)	0.017
	QTP + V	28.89	13.50	−16.25 (−24.93, −7.57)	<0.001
**CUDOS**	time * group				0.640
	QTP + L	14.11	10.29	−5.41 (−10.98, 0.16)	0.071
	QTP + V	10.50	6.00	−1.84 (−8.7, 5.02)	0.605
**YMRS**	time * group				0.300
	QTP + L	11.33	7.43	−4.82 (−9.33, −0.31)	0.047
	QTP + V	7.20	5.00	0.23 (−5.36, 5.82)	0.936

^#^: QTP + V: quetiapine plus sodium valproate group; QTP + L: quetiapine plus lithium carbonate group; analyses were conducted with the use of a generalized linear mixed model, treating group as fixed effect, patient ID as random effect, time * group as interaction and age, sex as explanatory variables.

## Data Availability

Data is contained within the article.

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
