# Peer review of "Efficacy of Quetiapine Monotherapy and Combination Therapy for Patients with Bipolar Depression with Mixed Features: A Randomized Controlled Pilot Study"

_pharmaceuticals, 2023, doi:10.3390/ph16020287_

Round 1
Reviewer 1 Report
In addition to a general review in English, the article would benefit from a better theoretical description in the introduction.
The methods also need clarification. Also, it is not clear how the participants were selected, nor the type of population (inpatient vs outpatient, patients with a first episode of illness or already with chronic illness).
Reviewer 2 Report
This paper aims to address an important clinical issue in the pharmacotherapy of bipolar depression with mixed features defined by DSM-5, as no agents have sufficient evidence for first line treatment according to updated guidelines. They conducted a randomized controlled pilot study to evaluate the effectiveness of quetiapine (QTP) monotherapy in patients with DSM-5 bipolar depression with mixed feature, and compared the combination therapy of QTP plus valproate versus QTP plus lithium for those patients responded insufficiently to QTP. The study was well-designed and the results provided new evidence for the pharmacotherapy of DSM-5 bipolar depression with mixed feature.
1.Title: I suppose the title should be more simplified.
2.Patients: How about the medication status of patients at the time of enrollment? Please add more details to the inclusion and exclusion criteria.
3.Did you measure the plasma levels of valproate/lithium?If did, please add the information; if not, better mention it in the limitation part.
4.2 weeks’ observation of QTP is inadequate even by 20% MARDS score reduction. Time effect of V or L or neither should be considered in the results within the responders and the nonresponders.
5.“Chi-squared tests or Fisher’s exact tests were adopted for group comparison of categorical variables.” However, you didn’t use them in Table 1.
6.Description of the Scales was missing. “with a score ≥20 on the MADRS and a YMRS score of ≤19 at both screening and baseline”. The reason is necessary and how many BD-depression patients were excluded by these exclusion criteria.
7.“In table 1, the mean (±SD) age, education and duration of the participants was 21.80±7.05, 12.33±2.34 and 63.41±55.41, respectively.” It was supposed that the duration was calculated by month. Please clarify it. The medication dose was not listed.
8.In figure 1, “treatment” was misspelled at the last line. Moreover, were there 36 patients in the QTP group? Then, only 23 patients had finished the study after 8 weeks (41%, 21/56), with a high exit rate.
9.Figure 1 cannot be observed at this review stage.
10.The rebuild of the sequence of the Change of the scores section is required. At first you should show if there exists an interaction effect (time*group) and then the main effect of what were within groups or between groups. Every result ought to be described ahead of a p value. The results of go downs or creep ups was an exact data, rather than simply by observation.
11.In the conclusion section, you discussed low serum lithium concentration is associated with anxiety symptoms. Was dose of lithium/valproate related to your study?
Reviewer 3 Report
ABSTRACT
Recommend following BOMRC format to simplify readability for the original study
INTRODUCTION
Line 62 needs clarification as CANMAT/ISBD 2018 had shown some possible evidence with certain SGA.
METHODS
Typo error: methods need to be section 2 (instead of current) and then followed by results.
Recommend making a linear progress graph for content in subsection 2.1. I see Figure 1 but it hasn't been cited in subsection 2.1
RESULTS/DISCUSSION
Very good work was done by the authors in summarizing results that align with objectives. Recommend to add subsection of clinical implication.
